# Whole Page Unbiased Learning to Ranking

## ABSTRACT

The page presentation biases in the information retrieval system, especially on the click behavior, is a well-known challenge that hinders improving ranking models' performance with implicit user feedback. Unbiased Learning to Rank (ULTR) algorithms are then proposed to learn an unbiased ranking model with biased click data. However, most existing algorithms are specifically designed to mitigate position-related bias, e.g., trust bias, without considering biases induced by other features in search result page presentation(SERP), e.g. attractive bias induced by the multimedia. Unfortunately, those biases widely exist in industrial systems and may lead to an unsatisfactory search experience. Therefore, we introduce a new problem, i.e., whole-page Unbiased Learning to Rank(WP-ULTR), aiming to handle biases induced by whole-page SERP features simultaneously. It presents tremendous challenges: **(1)** a suitable user behavior model (user behavior hypothesis) can be hard to find; and **(2)** complex biases cannot be handled by existing algorithms. To address the above challenges, we propose a Bias Agnostic whole-page unbiased Learning to rank algorithm, named BAL, to automatically find the user behavior model with causal discovery and mitigate the biases induced by multiple SERP features with no specific design. Experimental results on a real-world dataset verify the effectiveness of the BAL.

## CCS CONCEPTS

• **Information systems** → **Social networks**; **Learning to rank**.

## KEYWORDS

Information Retrieval, Unbiased Learning to Rank, User Modeling

**ACM Reference Format:**
Anonymous Author(s). 2018. Whole Page Unbiased Learning to Ranking. In *Woodstock '18: ACM Symposium on Neural Gaze Detection, June 03–05, 2018, Woodstock, NY*. ACM, New York, NY, USA, 10 pages. https://doi.org/10.1145/1122445.1122456

## 1 INTRODUCTION

A key component in modern information retrieval systems is the ranking model which provides highly relevant documents given the particular user query. It is of great practical value in multiple scenarios, e.g., e-commerce and the web search engine. Ideally, the ranking model should be learned with experts' annotated relevance labels, which, unfortunately, is both expensive and labour-intensive. A more affordable alternative is to utilize the implicit biased user

feedback, i.e., click, as training labels, which are easy to collect and more suitable to meet the large data requirement in deep learning. However, the click is affected by biases induced by search result presentation features (SERP) from the whole page, such as trust bias [1] induced by position, attractive bias [44] induced by abstract, and vertical bias [9, 25, 27] induced by multimedia types. To mitigate biases in implicit user feedback, Unbiased Learning to Rank (ULTR) [23, 39, 40] algorithms have been introduced. Generally, most existing ULTR algorithms often consist of two procedures: **(1)** User behavior model design (i.e., the user behavior assumption [11, 28, 32]) aims to model how the user click behavior is influenced by both SERP features and relevance score, such as the trust position-based model [1], and the mobile user behavior model [28]. Unfortunately, these user behavior models are proposed with predefined assumptions from human knowledge, which are designed for position-related bias and inapplicable to biases from more complex SERP features. **(2)** Unbiased learning focuses on mitigating biases and learning toward an unbiased ranking model under the defined user behavior model, e.g., the IPW [33] method. However, existing ULTR algorithms only consider position-related biases. Such position-based Unbiased Learning to Rank (PB-ULTR) is insufficient for the modern search engine that often presents more biases on multiple SERP features. Consequently, unsatisfactory results have been observed on the real-world dataset [49], where PB-ULTR solutions, including IPW [23], PairD [18], and REM [40], perform no better than that trained on biased click data directly.

The challenges mentioned above motivate us to introduce a new research problem in this work, i.e., whole-page Unbiased Learning to Rank (WP-ULTR). It aims to simultaneously mitigate all biases introduced by features on the **whole** search **page** presentation, such as position, SERP height, and multi-media type. Unfortunately, there are tremendous challenges in user behavior model design and unbiased learning for WP-ULTR. For the **user behavior model design** step, heuristic user behavior hypothesis in PB-ULTR is not applicable for the following reasons: **(1)** Existing user behavior hypotheses have been designed specifically for the position, which cannot be directly extended to other SERP features. **(2)** Crafting the user behavior model under the effects of multiple SERP features is difficult. For example, the web with video usually has larger height, often leading to more clicks. However, such bias has not been touched by existing literature. **(3)** An universal user behavior hypothesis, e.g., the examination hypothesis [32], might not exist in the WP-ULTR. Different scenarios may have distinct SERP features, and each scenario typically requires a specific user behavior model. For example, a vertical search engine like Linkedin[1] has a substantially different whole-page presentation from a general search engine like Google[2].

We are desired to design an algorithm that can automatically identify the complicated relationships in the user behavior model instead of heuristic designs. For the **unbiased learning** step, the

---

[1]https://www.linkedin.com
[2]https://www.google.com

major challenge attributes to the complexity of the user behavior model. Existing unbiased learning algorithms have usually been designed for the specific and simple user behavior model on the position. However, biases of WP-ULTR are much more complicated and beyond the scope of existing PB-ULTR algorithms. For example, existing algorithms do not consider the bias caused by **confounders** (i.e., a variable that causes spurious association between two target variables by influencing both of them) between click and the SERP features. In our WP-ULTR scenario, the relevance between query and document can be the confounder that introduces spurious association between the click and SERP features (e.g., SERP height, ranking position) by influencing both of them. Correspondingly, ignoring such bias may lead to inaccurate bias estimation and unsatisfying performance. Therefore, we need to design a more general unbiased learning algorithm suitable for general user behavior models.

In this work, we propose **BAL**, a **B**ias **A**gnostic whole page unbiased **L**earning to rank algorithm. It can automatically discover and mitigate biases induced by whole-page SERP features. For the **user behavior model design** step, instead of the hand-drafted heuristic design with the causal discovery algorithm, BAL can automatically design the user behavior model. We first introduce causal discovery techniques in the context of ULTR to discover the causal relationship among click, true relevance, and whole-page SERP features with a certain guarantee. Therefore, it avoids the difficulties to design the user behavior model with heuristic hypotheses which could be both inaccurate and labor-intensive. For the **unbiased learning** step, we propose a general unbiased learning algorithm that could easily handle various scenarios with different biases. We first recognize confounding effects on the user behavior model and propose an Importance Reweighting [26] based algorithm to remove the confounding affect. After that, we can estimate how SERP features and query-document relevance influence the user behavior, separately. Then we directly learn the effect of query-document relevance on the user behavior while ignoring the one from SERP features. Then, we further mitigate the bias induced by SERP features by blocking the gradient related to SERP features and only learning the effect from the relevance score to click directly. The advantages over the existing algorithms are two-fold. First, it can mitigate the relevance score's confounding bias on click and SERP features, i.e., the inaccurate bias estimation. Second, it can be easily extended to mitigate biases from multiple SERP features rather than only position. Remarkably, though the BAL does not rely on any predefined click behavior hypothesis, BAL is still explainable since it generates the causal graph to indicate how the click is biased on SERP features (depicted in Figure 3). In summary, the main contributions of our work are as follows:

- We introduce a novel problem whole-page Unbiased Learning to Rank (WP-ULTR) that considers biases induced by SERP features other than position.
- We propose a bias-agnostic whole-page ULTR algorithm BAL that can discover the user behavior model automatically and mitigate biases from multiple SERP features simultaneously.
- Extensive experiments on a large-scale real-world dataset indicate the effectiveness of our algorithm.

## 2 PRELIMINARY

In this section, we provide a brief introduction to the problem of whole-page unbiased learning to rank. The causal graph is then introduced to describe biases formally.

**Learning to Rank** The task of a ranking model $\hat{\mathbf{r}} = f(\mathbf{q}, \mathbf{d}; \Theta)$ is to rank the document with more relevance to the top of the ranking list. $\Theta$ is the model parameters. The query $\mathbf{q}$ is sampled from a collection of queries $Q$. Each query $\mathbf{q}$ is relevant to a number of documents $\mathcal{D}_{\mathbf{q}} = \{\mathbf{d}_i\}_{i=1}^{N}$ retrieved from all indexed documents $\mathcal{D}$. With the estimated relevance score $\hat{\mathbf{r}}$ on query and document, a ranking list $\pi_{\mathbf{f},\mathbf{q}}$ is generated by descendingly sorting documents $\mathcal{D}_{\mathbf{q}}$ according to $\hat{\mathbf{r}}$. Therefore, the goal of learning a rank model is to maximize the order-consistent evaluation metric i.e. $\vartheta$ (e.g., DCG [20], PNR [50], and ERR [7]) as

$$\mathbf{f}^* = \max_{\mathbf{f}} \mathbb{E}_{\mathbf{q} \in Q} \vartheta(\mathcal{Y}_{\mathbf{q}}, \pi_{\mathbf{f},\mathbf{q}}). \tag{1}$$

where $\mathcal{Y}_{\mathbf{q}} = \{y_{\mathbf{d}}\}_{\mathbf{d} \in \mathcal{D}_{\mathbf{q}}}$ is a set of annotated relevance labels $y_{\mathbf{d}}$ corresponding to $\mathbf{q}, \mathbf{d}$. Usually, $\mathbf{y}_{\mathbf{d}}$ is the graded relevance in 0-4 ratings, which indicates the relevance of document $\mathbf{d}_i$ as {**bad**, **fair**, **good**, **excellent**, **perfect**}, respectively. To learn the scoring function $\mathbf{f}$, a loss function is to approximate $\mathbf{y}_{\mathbf{d}}$ with $\mathbf{f}(\mathbf{q}, \mathbf{d})$ as

$$\ell(\mathbf{f}_{ideal}) = \mathbb{E}_{\mathbf{q} \in Q} \left[ \sum_{\mathbf{d} \in \mathcal{D}_{\mathbf{q}}} \Delta(\mathbf{f}(\mathbf{q}, \mathbf{d}), \mathbf{y}_{\mathbf{d}}) \right], \tag{2}$$

where $\Delta$ is a function that computes the individual loss for each document. However, the above loss is impractical since the annotated relevance labels acquired from expert judgment are expensive.

**Whole-Page Unbiased Learning to Rank** Unbiased Learning to Rank is proposed as an alternative and intuitive approach to utilize the user's implicit feedback as the training signal, which is much easier to obtain. For example, by replacing the relevance label $y_d$ with click label $c_d$ in Eq. 2, a naive empirical ranking loss is derived as follows

$$\ell_{naive}(\mathbf{f}) = \frac{1}{|Q_o|} \sum_{\mathbf{q} \in Q_o} \left[ \sum_{\mathbf{d} \in \mathcal{D}_{\mathbf{q}}} \Delta(\mathbf{f}(\mathbf{q}, \mathbf{d}), \mathbf{c}_{\mathbf{d}}) \right], \tag{3}$$

where $Q_o$ is the training query set. $\mathbf{c}_{\mathbf{d}}$, indicating whether the document $\mathbf{d}$ in the ranked list is clicked, is utilized as the training signal. However, the above loss function is biased on the whole-page SERP features $\mathbf{x} = \{\mathbf{x}_1, \mathbf{x}_2, \cdots, \mathbf{x}_n\}$ ,e.g., visual appearances, multimedia types, and position, related with the document $\mathbf{d}$. To address this issue, we propose whole-page Unbiased Learning to rank (WP-ULTR). It aims to find the suitable loss function to remove the effect of data bias induced by multiple SERP features $\mathbf{x} = \{\mathbf{x}_1, \mathbf{x}_2, \cdots, \mathbf{x}_n\}$. Remarkably, $\mathbf{x}$ can be with arbitrary data types, e.g., categorical, continuous, and ordinal features.

**Bias** Biases in ULTR domain could be described as the difference between click user behavior and query-document relevance. User behavior model, which describes how click $c$ is influenced by the query-document relevance $e$ and SERP features $x$, is generally utilized to provide a formal description on biases. For example, Position-based user behavior Model [23] assumes that a document displayed at a higher position is more likely to receive clicks than a document at a lower position. It suggests that bias is from the

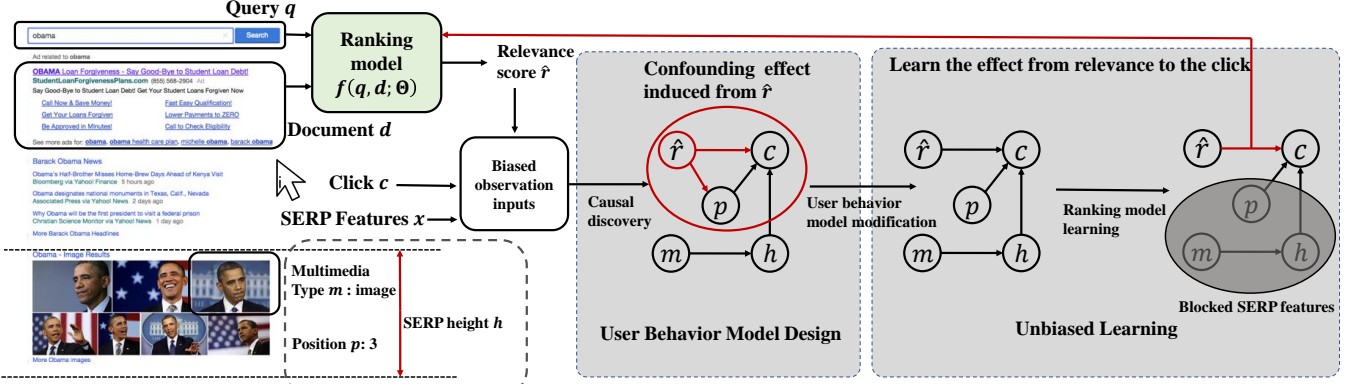

Figure 1: An overall procedure illustration of the BAL algorithm. The biased observation inputs include the query-document relevance score r̂, the click c, and SERP features x. BAL has two steps: (1) User behavior model design step learns a causal graph with causal discovery; (2) Unbiased learning steps mitigate biases found in the causal graph towards an unbiased model.

influence from the position. ULTR algorithms are then designed to mitigate biases in the user behavior model.

Existing user behavior models [1, 23] are hand-crafted assumptions on how the user behavior is affected by the position SERP feature. However, existing user behavior models may not meet the real-world scenario for they focus on the position-based biases, while ignoring many other biases induced by other SERP features as shown in [49]. A more detailed description on the potential biases in the ranking system can be found in [5]. Moreover, existing user behavior models can be hard to extend to other SERP features with different data types and unique properties. The main obstacle is the lack of suitable heuristic design on how SERP features affect user behavior. To address the above challenge, we adopt the causal graph to describe the user behavior model. The advantages are as follows: **(1)** The causal graph can be easily extended to more SERP features. It can denote the complex relationships among click, relevance score, and multiple SERP features; and **(2)** Causal discovery techniques, instead of the traditional heuristically designed user behavior model, can be employed to discover biases with causal graphs automatically. In the following paragraph, we will discuss the details on defining bias and unbiasedness in causal graph.

**Causal graph** The causal graph $\mathcal{G} = (\mathbf{V}, \mathbf{E})$ is utilized to describe the user behavior model. $\mathbf{V}$ is a set of nodes where each node corresponds to a variable. In WP-ULTR, variables include those corresponding to SERP features $x = \{x_1, x_2, \cdots, x_n\}$, user click $c$, and the query-document relevance score $\hat{r}$ generated by the ranking model. $\mathbf{E}$ is a set of directed edges where each edge denotes the causal relationship between cause (starting node) and effect (ending node). Bias on WP-ULTR can be described as the causal effect from SERP features $x$ to click $c$. Unbiasedness is defined as precisely estimating and learning the causal effect from relevance score $\hat{r}$ to click. An example can be found in Fig. 2(a)(3).

The confounding effect is the main obstacle on estimating the causal effect from SERP features $x$ to click $c$. **Confounding effect** means that two variables are dependent on a hidden variable, i.e., a confounding variable. It appears as a spurious correlation between those two variables. Not surprisingly, there exists such confounding effect in WP-ULTR. An example is illustrated in Fig. 2(a)(1). The click $c$ and the position $p$ are both dependent on the relevance score

$\hat{r}$. Without explicitly considering these hidden causal relations, the spurious correlation between position and click could greatly bias ranking models. Unfortunately, most of the current developments have not considered this confounding bias. To mitigate such confounding effect, we block the backdoor path from relevance score $\hat{r}$ to the position. We can then achieve a correct estimation on the causal effect from SERP features $x$ to click $c$ as shown in Fig. 2(a)(2). Learning the causal effect leads to an unbiased ranking system.

## 3 BIAS AGNOSTIC LEARNING ALGORITHM

In this section, we first provide an overview of our BAL algorithm shown in Figure 1. It consists of two major steps: the user behavior model design step and the unbiased learning step.

The user behavior model design step aims to identify and estimate biases from logged user behavior data. Specifically, we first utilize the data-driven causal discovery technique to automatically find the user behavior model. The user behavior model is described with the causal graph, which shows the complicated relationship among click $c$, relevance score $\hat{r}$, and SERP features $\mathbf{x} = \{x_1, x_2, \cdots, x_n\}$. An example is illustrated in Fig. 1, where we could identify the bias effect of SERP features, including multimedia type, SERP height, and ranking position. Moreover, relevance score $\hat{r}$ shows a typical confounding effect on $p$ and $c$. An influence score estimation is utilized to estimate how click is affected by different SERP features.

With biases found in the above step, the unbiased learning step aims to mitigate biases and learn an unbiased ranking model in two sub-steps: user behavior model modification and ranking model learning. In the former sub-step, we mitigate the confounding effect $\hat{r}$ via removing the backdoor path from $\hat{r}$ to $p$ with sample reweighting. In the ranking model learning sub-step, we achieve an unbiased ranking model by only learning the effect of relevance score to click by blocking the gradient from other SERP features.

## 3.1 User Behavior Model Design

In this subsection, we focus on utilizing the causal discovery algorithm to automatically find the user behavior model with the biased observation data. Then we can identify biases in the user behavior model and estimate the influence score on the SERP feature for measuring and mitigating biases in the unbiased learning steps.

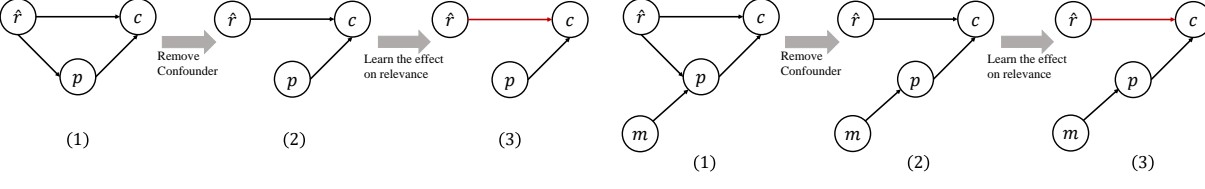

(a) The confounder case1 where $\hat{r}$ is the only parent node for $p$. $p$ is the SERP feature related to click $c$

(b) The confounder case2 where both $\hat{r}$ and other SERP feature $m$ are parent nodes for $p$. $p$ is the SERP feature related to the click $c$.

**Figure 2: Two representative cases on how the relevance score $\hat{r}$ reveals the confounding bias. The unbiased learning algorithm first removes the confounding bias then learns the click effect on the relevance score. $\hat{r}$, $r$, $c$, $p$, and $m$ correspond to the relevance score, true relevance, click, position, and multimedia types, respectively.**

**SERP feature preprocessing** The WP-ULTR problem has more SERP features with multiple data types, which PB-ULTR algorithms cannot handle. In general, there are three SERP feature types: continuous, ordinal, and categorical. Since deep learning is more suitable for continuous features, we convert ordinal and categorical features into continuous features. For the ordinal feature like the ranking position $\mathbf{p}$, we transform it into a continuous one with a Bradly-Terry model [6, 29, 41, 42] as $p_{i,j} = \frac{e^{s_i}}{e^{s_i}+e^{s_j}}$, where $p_{i,j}$ is the probability that the ranking position $i$ is lower than the position $j$. $s$ is the transformed continuous score, where a larger score represents a higher rank. Then we can obtain the continuous score $s$ by maximizing the likelihood optimization objective shown below. With all the position pairs $(i, j) \in \varpi$ satisfying position $i$ lower than position $j$, we maximize the log-likelihood of $\varpi$ as $\sum_{(i,j)\in\varpi} \log p_{i,j}$. For example, the score $s$ on position $\mathbf{p} = 2$ should be larger than any score with a higher position $\mathbf{p} > 2$.

For the categorical features, e.g., multimedia type, we utilize the embedding $\mathbf{E} = \{\mathbf{e}_1, \mathbf{e}_2, \cdots, \mathbf{e}_n\} \in \mathbb{R}^{n \times d}$ to transform the discrete one into the learnable continuous embedding.

**Causal discovery** The fundamental challenge in WP-ULTR is that multiple SERP features are newly introduced. It adds more difficulties to heuristically designing a suitable user behavior model. To address this challenge, we adopt the causal discovery technique, which can automatically find the user behavior model with the training data. It identifies the causal relation among the click $\mathbf{c}$, whole-page presentation features $\mathbf{x} = \{\mathbf{x}_1, \cdots, \mathbf{x}_n\}$, and the relevance score $\hat{\mathbf{r}}$ generated by the biased ranking model. Specifically, we apply the PC algorithm [35] with Kernel-based Conditional Independence test (KCI) [46] to discover the causal graph with mixed-type data. PC algorithm consists of two phases, i,e, skeleton search and orientation propagation. In the first phase, edges are removed from a complete graph in a certain order by the results of (conditional) independence tests; in the second phase, orientations are decided by a set of rules. KCI is a conditional independence test method which could consider both continuous and discrete input data. It is a kernel-based non-parametric method with the guarantee to apply to general distributions without limitations on functional relations and data types [46]. We refer to [35, 46] for more details. By applying the considered causal discovery method, we aim to recover the hidden structure in the context of real-world learning-to-rank. Moreover, some prior knowledge in ULTR is predefined on the causal graph as follows: **(1)** relevance score $\hat{r}$ must

be the parent of click $c$ since users only click the document when it is relevant to the query. **(2)** click $c$ cannot be the parent of SERP features $x$ since users click the document after they observe the SERP features. **(3)** SERP features $x$ cannot be the parent of relevance score $\hat{r}$. The SERP features are designed to optimize the ranking metric, for example, DCG [20], after we estimate the relevance of all candidate documents. Therefore, the SERP features are determined by the relevance score to some extent, and SERP features are not the parent of the relevance score.

The causal graph $\hat{\mathcal{G}}$ generated by the causal discovery algorithm is utilized to describe the user behavior model. It can help identify biases in user behaviors easily. Based on the aforementioned prior knowledge, there are two types of potential bias as follows: **(1)** Confounding bias: the relevance score $\hat{r}$ is connected with both the SERP feature $x_i$ and the click $c$. Meanwhile, the SERP feature $x_i$ is connected with click $c$. Then the spurious correlation exists from $x_i$ to $c_i$, which can lead to the overestimation on the relationship between $x_i$ and $c_i$. **(2)** SERP bias: both SERP features $x_i$ and the relevance score $\hat{r}$ are connected with the click $c$. Then if we learn directly from the click with the loss in Eq. (3), the model will learn the effects from both the SERP feature $x_i$ and the relevance score $\hat{r}$ to click. However, the ranking model should only consider the relevance between the query and the document with no effect from the SERP feature $x_i$.

**Influence score estimation on the SERP feature** After the causal discovery sub-step, we can identify where biases are from. However, how those biases affect the click behavior for each data instance is still unclear. To achieve this goal, it is necessary to first estimate the influence on the SERP feature. The influence score estimation on the SERP feature is defined as $p(x_i|\pi_{x_i})$, the conditional probability of the SERP feature $x_i$ given its parent variables $\pi_{x_i}$ including the relevance score and other SERP features connected in the causal graph. Intuitively speaking, the influence score estimates the sample influence on the confounding bias, i.e., changes of user behavior model whether the confounding bias is mitigated or not.

Multiple layer perceptron (MLP) is utilized as the probability estimator. It is expressive to deal with most parametric families of conditional probability distributions $P(x_i|\pi_{x_i})$. For each SERP feature $\mathbf{x}_i$, we learn an MLP parameterized with $\phi_i = \{\mathbf{W}_i^1, \mathbf{W}_i^2, \cdots, \mathbf{W}_i^L\}$, where $L$ is the number of layers in MLP. We first concatenate click $\mathbf{c}$, relevance score $\hat{\mathbf{r}}$, and SERP features $\mathbf{x}$ as the input data, $\mathbf{Z} = [\mathbf{c}, \hat{\mathbf{r}}, \mathbf{x}_1, \cdots, \mathbf{x}_n] \in \mathbb{R}^d$. Then we preprocess each SERP feature

$\mathbf{x}_i$ to the corresponding masked input $\mathbf{Z}_{\pi_{x_i}}$ by only keeping features corresponding to the parent nodes of $\mathbf{x}_i$ while masking others. Then, we forward the masked input to the corresponding network as follows:

$$\mathbf{H}_i = \mathbf{W}_i^L f(\cdots f(\mathbf{W}_i^2 f(\mathbf{W}_i^1 \mathbf{Z}_{\phi_i})) \cdots) \tag{4}$$

where $\mathbf{H}_i \in \mathbb{R}^m$ with $m$ parameters which describe the desired distribution. For example, if the desired distribution is a Gaussian distribution, then $m = 2$ corresponds to the mean and standard deviation, respectively.

The maximum likelihood optimization objective is as follows:

$$\max_\phi \mathbb{E}_{X \sim P_X} \sum_{i=1}^{|\mathcal{G}|} \log p_i \left( \mathbf{X}_i \mid \mathbf{X}_{\pi_i}; \phi_i \right) \tag{5}$$

Remarkably, the maximum likelihood objective may be different for different types of SERP features. In this work, we use polynomial distribution for categorical features and Gaussian distributions for continuous ones, respectively.

## 3.2 Unbiased learning

The goal of the unbiased learning step is to mitigate biases and learn toward an unbiased ranking model. In this work, we propose following two sub-steps to mitigate the biases, i.e., the confounding bias and SERP bias, discussed in Section 3.1: **(1)** Unbiased user behavior model modification sub-step aims to diminish the influence of confounding bias. It modifies the user behavior model for removing the backdoor path the relevance score to the SERP features with importance reweighting. **(2)** Ranking model learning sub-step aims to mitigate the SERP bias and learning towards a ranking model based on query-document relevance with no influence by SERP features. To achieve this goal, we blocks the gradients related to SERP features to avoid learning SERP features' affect on click. In the following, we first introduce the backbone model of WP-ULTR and then present the detail of the above two sub-steps.

**Ranking model backbone** The large-scale language model BERT architecture[14] is utilized as the default backbone. Query, document title, and document abstract are concatenated with [SEP] separator as the input. The ranking model is first warmed up by pretraining with both Masked Language Model (MLM) loss [14], and the naive loss in Eq. (3) with no ULTR techniques. Remarkably, the pre-trained ranking model is biased on SERP features. Then, our BAL algorithm is proposed to discover and mitigate biases in the pre-trained ranking model to train toward the unbiased one.

**Unbiased user behavior model modification** This subsection focuses on the confounding bias induced by the relevance score on the SERP feature and the click. As shown in Figure 2(a)(1), relevance score $\hat{\mathbf{r}}$ shows influence on both the position $\mathbf{p}$ and click $\mathbf{c}$. This bias induced by the confounder $\hat{\mathbf{r}}$ results in the spurious correlation between the position $\mathbf{p}$ and click $\mathbf{c}$. This spurious correlation leads to the SERP feature's over-estimated influence on the click and relevance's under-estimated influence on the click. Ideally, the causal graph structure without confounding effect is demonstrated in Figure 2(a)(2) where relevance $\hat{\mathbf{r}}$ has no relationship with the position $\mathbf{p}$. With this user behavior model, we can further accurately estimate the relevance's effect on the click and achieve the unbiased learning.

To remove the backdoor path from the relevance score to the SERP feature, we create a pseudo population following the unconfounded causal graph through importance reweighting [26]. Specifically, we explain two typical scenarios in Figure 2(a) and Figure 2(b). For the formal case, we identify the suitable reweighting score by making a comparison between the data distribution of the original causal graph $\hat{\mathcal{G}}$ in Figure 2(a)(1) and the expected causal graph $\mathcal{G}$ in Figure 2(a)(2). The data distribution for the original causal graph $\hat{\mathcal{G}}$ is:

$$p_{\hat{\mathcal{G}}}(\hat{\mathbf{r}}, \mathbf{c}, \mathbf{p}) = p(\hat{\mathbf{r}}) \cdot p(\mathbf{p}|\hat{\mathbf{r}}) \cdot p(\mathbf{c}|\mathbf{p}, \hat{\mathbf{r}}). \tag{6}$$

And the data distribution for the expected causal graph $\mathcal{G}$ is:

$$p_{\mathcal{G}}(\hat{\mathbf{r}}, \mathbf{c}, \mathbf{p}) = p(\hat{\mathbf{r}}) \cdot p(\mathbf{p}) \cdot p(\mathbf{c}|\mathbf{p}, \hat{\mathbf{r}}) \tag{7}$$

To reweight the data distribution of $\hat{\mathcal{G}}$ towards the data distribution of $\mathcal{G}$, the reweighting score on the position should be $\mathbf{w}_p = \frac{p(p)}{p(p|\hat{r})} = \frac{p_{\mathcal{G}}(\hat{r},c,p)}{p_{\hat{\mathcal{G}}}(\hat{r},c,p)}$, where $p(p|\hat{r})$ is the influence score on SERP feature from Section 3.1.

The other scenario is that the SERP feature $p$ is not only dependent on the click but also on other SERP features $m$ as illustrated in Figure 2(b). Similarly, the reweighting score is $\mathbf{w} = \frac{p_{\mathcal{G}}(\mathbf{r},c,p,\mathbf{m})}{p_{\hat{\mathcal{G}}}(\hat{\mathbf{r}},c,p,\mathbf{m})} = \frac{p(\mathbf{p}|\hat{\mathbf{r}},\mathbf{m})}{p(\mathbf{p}|\mathbf{m})}$, where $p(\mathbf{p}|\hat{\mathbf{r}}, \mathbf{m})$ can be obtained from the SERP propensity score estimator easily. $p(\mathbf{p}|\mathbf{m})$ can be estimated via data permutation in batch as $p(\mathbf{p}|\mathbf{m}) = \sum_{\hat{\mathbf{r}}' \in \mathbf{R}} p(\hat{\mathbf{r}}') p(\mathbf{p}|\hat{\mathbf{r}}', \mathbf{m})$. $\mathbf{R}$ is a batch of relevance score data.

Taking the above two scenarios into consideration, we can remove the backdoor path on each SERP feature. Then, the effect of SERP features and relevance score on the click can be corrected by maximizing the reweighted maximum likelihood objective as:

$$\max_\phi \mathbb{E}_{X \sim P_X} \mathbf{w}_i \log p \left( \mathbf{c} \mid \mathbf{Z}_{\pi_c}, \phi_c \right) \tag{8}$$

where $\mathbf{w} = \prod_i \mathbf{w}_i$ is the total reweighting score defined as the product of the reweighting score $\mathbf{w}_i$ on each SERP feature with confounding bias on the relevance score. $p \left( \mathbf{c}_i \mid \mathbf{Z}_{\pi_c}; \phi_c \right)$ can be estimated with an MLP parameterized with $\phi_c$, similar to the influence estimator on the SERP feature. Then, we can get a well-learned MLP estimator on the data distribution of the unconfounded causal graph $\mathcal{G}$.

**Ranking model learning** After mitigating the confounding bias from the relevance score, the SERP bias still exists, where the relevance score and the SERP feature can affect the click user behavior. The ranking model should only consider the relevance between the query and the document while ignoring SERP features. To learn a ranking model without effect from SERP features, we block the gradient related to SERP features and only learn the relationship from relevance to click, shown in Figure 2(a)(3) with a red arrow. We utilize the item-wise loss:

$$\max_\Theta \mathbb{E}_{X \sim P_X} \mathbf{w}_i \log p \left( \mathbf{c} \mid \mathbf{Z}_{\pi_c}, \phi_c \right) \tag{9}$$

where $\Theta$ is the parameters of the ranking model, which is BERT in our experiment. Note that the unbiased user behavior model modification sub-step utilizes the same optimization object but on a different parameter set, $\phi_c$, with the goal of removing the backdoor path. Ultimately, we discover and mitigate biases from multiple SERP features and learn towards an unbiased ranking model.

**Table 1: Performance comparison of BAL algorithm and PB-ULTR baseline algorithms on the Baidu-ULTR dataset. The best performance is highlighted in boldface. The first and second terms are performance expectation and standard deviation.**

|        | DCG@1           | DCG@3           | DCG@5           | DCG@10          | ERR@1           | ERR@3           | ERR@5           | ERR@10          |
|--------|-----------------|-----------------|-----------------|-----------------|-----------------|-----------------|-----------------|-----------------|
| Naive  | 1.235±0.029     | 2.743±0.072     | 3.889±0.087     | 6.170±0.124     | 0.077±0.002     | 0.133±0.003     | 0.156±0.003     | 0.178±0.003     |
| IPW    | 1.239±0.038     | 2.742±0.076     | 3.896±0.100     | 6.194±0.115     | 0.077±0.002     | 0.133±0.003     | 0.156±0.004     | 0.178±0.003     |
| REM    | 1.230±0.042     | 2.740±0.079     | 3.891±0.099     | 6.177±0.126     | 0.077±0.003     | 0.132±0.003     | 0.156±0.004     | 0.178±0.004     |
| PairD  | 1.243±0.037     | 2.760±0.078     | 3.910±0.092     | 6.214±0.114     | 0.078±0.002     | 0.133±0.003     | 0.156±0.003     | 0.179±0.003     |
| DLA    | 1.293±0.015     | 2.839±0.011     | 3.976±0.007     | 6.236±0.017     | 0.081±0.001     | 0.137±0.001     | 0.160±0.001     | 0.181±0.001     |
| **BAL** | **1.383±0.018** | **3.086±0.078** | **4.088±0.142** | **6.410±0.117** | **0.098±0.003** | **0.153±0.004** | **0.171±0.004** | **0.192±0.003** |

**Table 2: Performance comparison of ULTR algorithms on queries with different frequencies in the Baidu-ULTR dataset. The best performance is highlighted in boldface.**

|        | DCG@1 | | DCG@3 | | DCG@5 | | DCG@10 | |
|--------|-------|-------|-------|-------|-------|-------|--------|-------|
|        | High  | Tail  | High  | Tail  | High  | Tail  | High   | Tail  |
| Naive  | 1.730±0.053     | **0.875±0.044** | 3.726±0.052     | 2.028±0.072     | 5.029±0.093     | 3.060±0.086     | 8.873±0.118     | 4.203±0.107     |
| IPW    | 1.743±0.061     | 0.872±0.074     | 3.697±0.174     | 2.047±0.101     | 5.020±0.110     | 3.078±0.065     | 8.872±0.133     | 4.216±0.080     |
| REM    | 1.734±0.073     | 0.863±0.051     | 3.711±0.119     | 2.034±0.054     | 5.015±0.123     | 3.106±0.079     | 8.869±0.136     | 4.320±0.097     |
| PairD  | 1.752±0.068     | 0.873±0.064     | 3.674±0.122     | 2.060±0.084     | 4.938±0.181     | 3.129±0.073     | 8.898±0.141     | 4.261±0.127     |
| DLA    | 1.883±0.099     | 0.864±0.053     | 3.791±0.057     | **2.146±0.016** | 5.054±0.033     | **3.332±0.019** | 9.017±0.029     | 4.689±0.013     |
| BAL    | **2.100±0.113** | 0.861±0.107     | **4.525±0.083** | 2.039±0.113     | **5.734±0.129** | 2.890±0.132     | **9.247±0.157** | **4.710±0.175** |

## 4 EXPERIMENT

In this section, we design experiments to validate the effectiveness of BAL by answering the following research questions.

- **RQ1:** How does the performance of BAL the first algorithm in WP-ULTR, compare with the state-of-the-art PB-ULTR algorithms in the real-world dataset?
- **RQ2:** How BAL performs on queries with different frequencies?
- **RQ3:** How do different components in BAL affect effectiveness?
- **RQ4:** How does the user behavior model (the causal graph) change along with the ULTR procedure?

### 4.1 Experiment Setting

**Dataset.** Our experiments are conducted on the largest real-world ULTR dataset, *Baidu-ULTR* [49], to analyze the effectiveness of our proposed BAL algorithm. It includes 383,429,526 queries and 1,287,710,306 documents with clicks recorded for training. 7,008 queries and 367,262 documents with expert annotations are utilized for evaluation. The relevance annotation of each document to the specific query is judged by expert annotators in five levels, i.e., {bad, fair, good, excellent, perfect}. *Baidu-ULTR* is the only open-source ULTR dataset providing the whole-page SERP features, including the ranking position, the document title, the document abstract, the multimedia type of the SERP (e.g., video, image, and advertisement), the SERP height (the vertical pixels), and the maximum SERP height. The title and the abstract of the document are utilized to generate the query-document relevant score. And the click feedback is collected from the real-world Baidu search engine, rather than the synthetic data generated with the heuristic assumption in most existing ULTR datasets [7, 13, 31].

**Remark.** Synthetic experimental analysis on the previous PB-ULTR works [2, 19, 23, 40] cannot be utilized in the WP-ULTR scenario. The position-based click model (PBM) [8] is usually utilized to generate the synthetic click data. The PBM utilizes the

examination assumption [32] as $p(c) = p(o) \cdot p(r)$, where $p(c)$, $p(o)$, and $p(r)$ are the probabilities of click, observation, and relevance, respectively. However, PBM and other click models cannot be extended to the WP-ULTR scenario since the generated click data is only biased to the position while ignoring other SERP features. Moreover, we cannot examine how well the algorithm estimates the bias since we do not know what the true bias is in WP-ULTR.

**Metric.** We follow the same evaluation metrics utilizing in the Baidu-ULTR dataset [49] and the WSDM CUP 2023[3], including DCG (Discounted Cumulative Gain) [20] and ERR [16] (Expected Reciprocal Rank), to access the performance of the relevance model. For both metrics, we report the results at ranks of 1, 3, 5, and 10.

**Baselines & Implementation details.** We select representative ULTR algorithms as baseline methods including IPW [23], DLA [2], REM [40], and PairD [19]. Notice that all the algorithms which belong to the PB-ULTR algorithm only consider position-related biases. Moreover, a naive algorithm is utilized which trains the ranking model directly with the biased click data, as stated in Eq. 3. We adopt the open-source ULTR toolkit[4] [37] for both baseline methods implementation and hyperparameter selection. Remarkably, the naive algorithm may not be necessary to show the worst performance. Other ULTR baseline algorithms may not also perform as expected if real-world datasets do not meet their predefined user behavior hypotheses.

**Remark:** All the baseline methods are PB-ULTR methods, which only consider the position-based biases. They can still work well in our WP-ULTR scenario since position-related biases are still very important as shown in [49]. However, it can be hard to extend those baselines to mitigate biases induced by other SERP features. The main reasons are two-fold: **(1)** baseline algorithms are specifically

---

[3]https://aistudio.baidu.com/aistudio/competition/detail/534/0/introduction
[4]https://github.com/ULTR-Community/ULTRA_pytorch

designed for position-based user behavior models **(2)** SERP features can have different feature types with unique properties. It still remains under-explored how those SERP features biased the user behavior. There is no existing predefined user behavior model related to other SERP features other than position that we can directly apply baselines on. Therefore, we use the original baseline implementation which only mitigates the position-based biases

## 4.2 Overall Performance Comparison (RQ1)

The experimental results of all baselines, and our proposed BAL algorithm on DCG@{1, 3, 5, 10} and ERR@{1, 3, 5, 10} are illustrated in Table 1. We can observe that our BAL algorithm shows consistently better results than all baseline algorithms across all the evaluation metrics. The maximum gains on DCG@N metrics and ERR@N metrics over the best baseline performance are 0.247 and 0.017, respectively. Remarkably, maximum gains on DCG@N and ERR@N are from $N = 1, 3$, respectively. It indicates that our algorithm can be more beneficial on those small-screen devices like mobile phones where users usually only browse a few documents. For the baseline methods focusing on the PB-ULTR scenarios, PairD, REM, and IPW show similar performance with the naive algorithm while the DLA shows limited performance improvement. The existing PB-ULTR algorithms are not suitable for the real-world dataset with more complicated biases. It indicates the necessity of introducing the new WP-ULTR problem. Further fine-grain analysis on the effectiveness of BAL over existing PB-ULTR algorithms is in Section 4.4.

**Discussion.** There could be two key reasons why PB-ULTR algorithms show no better performance than the biased naive algorithm. First, position-based hypotheses are not sufficient in the real-world scenario since there are biases induced by other SERP features. For example, users are more likely to click on the document with the video type. Evidence from one empirical observation can further support this perspective – the training loss of PB-ULTR algorithms can vary from 100 to 0.5 between two training steps while the naive algorithm shows a more stable training procedure. This attributes to the unsuitable reweighting score on the inaccurate position hypothesis in the training procedure across different data batches. Second, position-based biases have been extensively studied thus existing PB-ULTR algorithms may have already been adopted by the existing search engines. That is why applying the same algorithm to the newly generated click data shows no additional benefit.

## 4.3 The Impact of Query frequency (RQ2)

In the real-world search engine, user search queries follow heavy-tailed distributions. We further conduct experiments to investigate how algorithms perform on queries with different frequencies. The evaluation set with expert annotations provides a frequency identifier for each query. Queries are split into 10 buckets descendingly according to the query frequency. We illustrate our performance on the queries with high and tail frequencies, which correspond to buckets 0, 1, 2, 3, 4, and buckets 5, 6, 7, 8, 9, respectively.

The experimental results are shown in Table 2. One observation is that all algorithms show much better performance on high-frequency queries than on tail-frequency queries. It indicates that learning to rank on the tail queries is still a challenging problem. Our proposed BAL algorithm can consistently perform significantly better than baseline algorithms for high-frequency queries. While

the performance on the tail-frequency queries is only slightly better than baseline methods. The main reason is that the statistic significance-based causal discovery algorithm is more likely to find biases in high-frequency queries. Another potential reason could be that the existing search results for tail queries are of low quality with unsatisfying search experiments. For example, there are 93% tail queries having no high-relevant documents with excellent or perfect relevant labels. The user behavior may become unstable without consistent causal patterns.

## 4.4 Effects of BAL and its components (RQ1,3,4)

In this subsection, we conduct further experiments to investigate the effectiveness of **(1)** the entire BAL algorithm, **(2)** the click model design, and **(3)** the unbiased learning. The experiment for the effectiveness of the unbiased learning step shows how the causal graph varies in different training steps, which also responds to RQ4.

**Effectiveness of BAL algorithm** To further support the effectiveness of BAL, we conduct a fine-grained analysis to see how algorithms perform on each original position. We first identified the document at each position given by the original ranker, i.e., the ranker trained with the naive algorithm. After re-rank by the ULTR algorithms, we calculate the new document average positions at the original position. Moreover, re-rank results with ground truth relevance annotation are also included as the upper bound. The results on *Baidu-ULTR* are illustrated in Figure 4(a). The curve and corresponding shadow area stand for the mean value and the standard deviation, respectively. Note that, we provide the results on the top 10 documents with the original ranker for better visualization. The top documents are also the most important for the ranking system. One observation is that the curves of position-based ULTR algorithms are more distant from that of true relevance labels. This further indicates position bias may have already been alleviated in the new click data. Contrastively, the curve of our BAL is the closest one to the true relevance annotation in most cases, which further indicates the superiority of our algorithm.

**Effectiveness of click model design** To verify the effectiveness of our click model design, we propose two variants of our proposed BAL without the causal discovery algorithm as follows: **(1)** Position Based BAL (PB-BAL) replaces the causal discovery algorithm with the predefined user behavior model frequently utilized in PB-ULTR, with only two edges, from the position and relevant score to the click. **(2)** Fully biased BAL (FB-BAL) recognizes that the click is biased on all SERP features. The causal graph of the FB-BAL is that the relevance score $r$ will affect all SERP features, and all SERP features will affect the click.

Experimental results are illustrated in Figure 4(b). We note that both variants show performance drop which indicates the importance of click model design. For the PB-BAL algorithm, its performance is only comparable with the naive algorithm. It further indicates that only considering position-based biases is not sufficient for the WP-ULTR problem. For the FB-BAL algorithm, it shows the worst performance across all algorithms. This observation suggests that if mitigating the bias improperly, the true relationship may be removed by mistake, leading to unsatisfied results.

**Effectiveness of unbiased learning** To verify how well unbiased learning works, we examine how the user behavior model (causal graph) changes along with the training procedure. This experiment

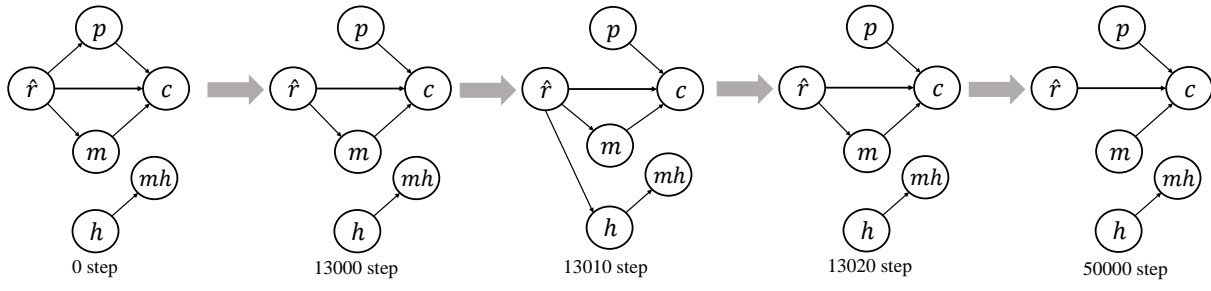

Figure 3: An illustration of how the causal graph changes during the training procedure on *Baidu-ULTR*. $r$, $c$, $p$, $m$, $h$, and $mh$ correspond to relevance score, click, position, multimedia types, SERP height, and maximum SERP height, respectively.

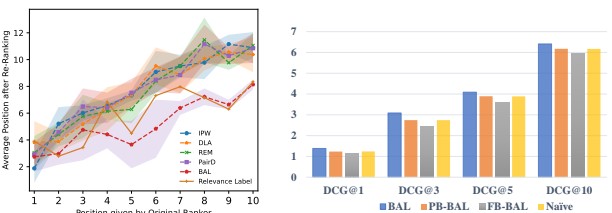

(a) Effectiveness of BAL algorithm    (b) Effectiveness of user behavior model design

Figure 4: (a) Average positions after re-ranking of documents at the top-10 original position by different ULTR methods. (b) Performance comparison on the naive algorithm, BAL and its variants without user behavior model design.

also provides a closer look at how biases are mitigated in the BAL training procedure. Results are shown in Figure 3. First, we figure out what the original causal graph and the expected graph after training are. For the original causal graph, we observe that multiple biases exist. The click is biased to both the multimedia type and the position. Remarkably, relationships also exist among SERP features. The height shows an effect on the maximum height. For the expected causal graph after training, all backdoor paths from the relevance score to SERP features are removed. Then we make the following observations on how the causal graph varies during training as follows: **(1)** The backdoor path from the relevance score to the position is mitigated first. This may further indicate that the position bias has already been alleviated by the feature process. It is much easier to mitigate position bias. **(2)** A new edge from relevance score to height appears. **(3)** The above new edge disappears quickly. The main reason for this observation is that the causal discovery algorithm is conducted on only a small portion of queries. Some relationships may appear in some particular data. However, it is not a common pattern across the whole dataset. It will quickly disappear in a few training steps. We ignore other similar small fluctuations that appeared in the training procedure. **(4)** The backdoor path from the relevance score to the position is removed. The above results verify the unbiased learning step of our BAL can successfully mitigate biases found by the click model.

## 5 RELATED WORK

**Unbiased Learning to Rank** Unbiased learning to rank aims to optimize with an unbiased estimation of the loss function based on biased user feedback. Existing ULTR algorithms only focus on position-based biases without taking other biases into consideration.

[39] mitigates the position bias with two key components: (1) an estimator on the predefined examination user behavior model [32] which can estimate the examination propensity score of each document by result randomization experiments. (2) an unbiased learning procedure with Inverse Propensity Weighting (IPW) [33]. [23] further proves that the learning objective with IPW is an unbiased estimate of the true relevance loss function based on the exam hypothesis. [40] improves the estimator with the regression-based EM algorithm with no requirement for the results permutation. [18] improves the unbiased learning procedure from a pointwise view to a pairwise view, which focuses more on the relative order of documents rather than the absolute relevant score of documents. Furthermore, [3] indicates that those two components are actually a dual problem and can be learned jointly. Despite the position bias, other position-based biases have also been taken into consideration. [1] takes the position-dependent trust bias [21] into consideration which is more robust to noise. However, those position-based algorithms cannot be successfully utilized in the real-world WP-ULTR scenario with biases induced by multiple SERP features. **User behavior model** The user behavior models [21, 22, 24, 32, 38, 44] are proposed to analyze how click is biased on the user SERP features. [32] finds the examination model where the model with a higher position is more likely to be examine. Then both position and examination will lead to the final click. [44] shows that the more attractive results are more likely to receive clicks. The most frequently utilized one is the [22] which proposes that the user shows more attention to the document with a higher position. In the existing studies, only position-related user behavior models are utilized for further study. More discussion and related work on **Click Model** and **Causal discovert** are in Appendix.

## 6 CONCLUSION & FUTURE WORK

In this paper, we propose a new problem, i.e., Whole-page Unbiased learning to Rank (WP-ULTR). It takes biases induced by all SERP features into consideration. In particular, we propose the BAL algorithm to automatically find and mitigate biases instead of the heuristic design. Extensive experiments on a large-scale real-world dataset indicate the effectiveness of BAL. One future direction is to extend our WP-ULTR to more user behaviors. As previous works only consider the click user behavior while other user behaviors, (e.g., SERP time, SERP count, and slip-off count), are ignored. We may be able to include more user behaviors to find more complex user behavior models and mitigate the real-world biases.

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

## A  EXPERIMENTAL DETAILS

We utilize the same backbone ranking model, which employs the BERT architecture, for approximating the scoring function $f(\mathbf{q}, \mathbf{d})$. The encoder of the ranking model has 12 layers, where each layer has 768 dimensions with 12 heads. It is pre-trained on both the MLM loss [14] and the naive loss shown in Eq. (3). We directly utilized the trained model parameters provided by [49], which can be found in the link[5]. An MLP decoder is built upon on the [CLS] embedding. The hidden layer dimensions of the 4-layer MLP decoders are 512-256-128. The library causal-learning[6] is utilized for the causal discovery for click model design. And the Adam optimizer is utilized for training. Each experiment result is obtained by 5 repeat runs on different random seeds. The best performance is selected based on the highest performance on DCG@10. All the models are trained on the machine with 80G Memory, 4 NVIDIA A100 GPUs, and 128T Disk.

## B  MORE RELATED WORKS

**Click Model** Instead of the end-to-end training in ULTR, click models [8, 12, 15, 43] mitigate the bias with the following two steps: (1) extract the true relevance feedback from the click data (2) use the new signal to train an unbiased ranking model. [12] is a cascade model designed with position bias. It assumes that users are more likely to click document in a higher position since users will read the result page from top to down. User behavior model [8] incorporates the examination bias based on the position and last click.

**Remark.** Click model is different from the user behavior model. The user behavior model is a probabilistic graph describing the existing biases in the search engine. By contrast, the click model and ULTR algorithms are specific techniques to utilize the assumptions in the user behavior model to learn an unbiased ranker.

**Causal discovery** Causal discovery aims to learn the causal relationship among variables from purely observational data. It serves as one of the key tools for scientific discovery in various fields, such as Biology [47]. One major class of methods, namely constraint-based methods, leverage conditional independence tests to estimate the causal graph (e.g., PC [35] and FCI [36]). Another popular category is score-based methods, of which the GES [10] is a representative one. Besides, much attention has been drawn to weakening the assumptions and extending the applicability, for instance, relaxing functional or distributional constraints [17, 34, 45, 48], focusing on mixed data types [4], or improving the scalability [30].

In our setting, we are required to deal with complex real-world data, suggesting that our method should be able to deal with general distributions with arbitrary functional relations and data types. By utilizing nonparametric tests, such as Kernel Conditional Independence test (KCI) [46], constraint-based methods (e.g., PC) can be applied in the general setting. Thus, we adopt PC with KCI for the procedure of structure learning.

---

[5]https://github.com/ChuXiaokai/baidu_ultr_dataset
[6]https://github.com/cmu-phil/causal-learn

