# OpenReview forum: "Whole Page Unbiased Learning to Rank"
_ACM.org/TheWebConf/2024/Conference — TheWebConf24 Oral_

### Official Review · Reviewer_wAnD · 2023-11-23

**Novelty:** 6
**Technical Quality:** 6

**Review:**

In information retrieval systems, user's implicit feedback, like clicks, is influenced by the presentation of the results. Thus, optimizing a system to clicks based on logged data naively will bias the trained model. The debiasing problem aims to fix this bias, and train an implicit-feedback optimized ranker that is not influenced by the original ranker. While most prior work on debiasing focuses on rank-based biases, the paper extends this framework to deal with general presentation biases. The paper suggests first learning a generic user modeling, and then utilizing this user modeling for debiasing. To validate the suggested approach, the paper learns an unbiased ranker on a large dataset, and shows that its ranking outperforms SOTA baselines on an annotated test set.

I believe that extending unbiasing beyond position-related bias is very important. As someone that comes from industry, I often see the need to take into account more presentation features.

I found the proposed method insightful. However, I found it difficult to follow all the details. In terms of writing, I would like to see more details about the ranking-model-learning phase, and more formal explanation of the user behavior model, in terms of structure, input and output. In addition, an important related work was only in the appendix. Please bring it to the main body, even at the cost of presenting RS2 or RS3 in the appendix.

I found the experimental part convincing and the results significant. I was happy to see that the evaluation wasn't based on a synthetic dataset, as common in debiasing works, and was made on a large ULTR dataset. Finally, I was curious to see how your system works compared to the baselines on a synthetic dataset where the user synthetically acts according to a well-known position-based logic. However, this is not in any way a condition to accept the paper.

**Questions:**

Can you provide more details about the ranking-model-learning phase, and more formal explanation of the user behavior model?

**Ethics Review Description:**

-

**Reviewer Confidence:**

3: The reviewer is confident but not certain that the evaluation is correct

**Scope:**

4: The work is relevant to the Web and to the track, and is of broad interest to the community

---

### Official Review · Reviewer_dLyR · 2023-11-23

**Novelty:** 3
**Technical Quality:** 3

**Review:**

#### [Summary]
This paper targets unbiased learning to rank problem. Pointing out that the existing papers are mostly specifically designed to mitigate position-related bias, this paper proposes a new problem, named Whole Page Unbiased Learning to Ranking (WP-ULTR), which encompasses various types of SERP features (e.g., multimedia features). This paper also proposes a new method, named Bias Agnostic whole page unbiased Learning to rank (BAL). The experiments are conducted on the Baidu-ULTR dataset.

#### [Strengths]
- This paper tackles an important research problem for the ranking model, and its unified debiasing strategy for various SERP features can be an incremental advancement over existing approaches.
- The proposed method is explained in detail and is easy to understand.
- This paper provides experimental results along with standard deviations.


#### [Weaknesses]
- The paper argues that introducing the WP-ULTR problem is one of its novel contributions. However, I expected a more formal and concrete definition of this problem. Currently, in Section 2, the paper only compares the ideal and naive loss, stating that "it aims to find a suitable loss function to eliminate data bias induced by multiple SERP features." Simply stating that the core distinction of this problem lies in the number of features makes it seem less important and novel. I believe a more concise definition and a plausible explanation are required to emphasize the significance of this problem.

- I have some concerns regarding the experiments in this paper. It appears that recent ULTR methods are not included in the baselines, and they are also not discussed in the related work. The most recent compared method was presented in 2019. I recommend that the authors discuss and compare more recent methods in the paper, such as [1] and [2], to evaluate the effectiveness of the proposed method.

     - [1] Can Clicks Be Both Labels and Features?: Unbiased Behavior Feature Collection and Uncertainty-aware Learning to Rank, SIGIR'22
     - [2] LBD: Decouple Relevance and Observation for Individual-Level Unbiased Learning to Rank, NeurIPS'22

**Questions:**

Please refer to the weaknesses section in my review. Thank you.

**Ethics Review Description:**

I don't have ethical concerns

**Reviewer Confidence:**

3: The reviewer is confident but not certain that the evaluation is correct

**Scope:**

4: The work is relevant to the Web and to the track, and is of broad interest to the community

---

### Official Review · Reviewer_fcH1 · 2023-11-24

**Novelty:** 5
**Technical Quality:** 3

**Review:**

The presented ideas are interesting and appear to be novel. The experiments are conducted on a very large real-life dataset. On the other hand, the submission has several major weaknesses:

1. The backbone ranking model is not very meaningful for this study: i) it relies on the bare BERT LLM and does not even use a fine-tuned relevance ranking model; ii) the relevance scores are computed based on the title and abstract, which appear on the SERP, and not the actual document content itself. Therefore, the experimental setting is not very realistic although it uses large-scale real-life data for evaluation.

2. The submission keeps talking about how result features (other than position) may introduce a click bias to motivate the reader about whole-page unbiased LTR. But, at the end of the study, it is unclear whether any of those features really introduced a click bias because this has never been evaluated. What fraction of the click bias is due to the position feature and what fraction is due to the remaining features (and what are they)? One possible experiment the authors can conduct is to fix the potential click bias due to all features in the causal graph (excluding the position feature) and see if this brings any improvement w.r.t. the naive baseline.

3. The results are hard to interpret and are not accompanied by take-away messages. As an example, according to Table 2, the performance is good on high-frequency queries, but weak on low-frequency queries, as somewhat expected. The submission does not give any detail about what "high frequency" really refers to, other than talking about some obscure notion of query buckets. If "high-frequency" queries refer to "head" queries, unfortunately, they are only a tiny fraction of unique queries and this raises questions about the impact of the attained results in a real-life setting.

4. The work is somewhat poorly presented. Many points are obscure or become clear much later after they are first mentioned in the text. Moreover, there are quite a few misleading statements. Examples:

- The user modeling part is mainly based on the following assumption: "relevance score r must be the parent of click c since users only click the document when it is relevant to the query": This assumption is simply incorrect. There are many examples where a user may click on an irrelevant document. Think about clicks due to misleading titles or nude images. True relevance is something that can be assessed after a click (i.e., after viewing the document), not before a click.

- Similarly, the assumption that "SERP features x cannot be the parent of relevance score r" holds only if the underlying ranking algorithm does not use any of the SERP features in x for unbiased training. Indeed, the authors later end up saying "PB-ULTR algorithms may have already been adopted by the existing search engines". Thus, it is better if the authors state that they assume the initial ranking algorithm does not involve any attempt for unbiased ranking.

- "the SERP features are determined by the relevance score to some extent": This is not always true either. A typical SERP includes results from multiple search verticals, each with its own ranking function and relevance score distribution. I guess the authors need to state that the scope of their work is limited to a single search vertical exposing results obtained through a single ranking model.

- The consistent use of "SERP features" is very confusing. The pointed features are "search result features", not "search engine result page features", i.e., they are associated with each "result snippet", not with the entire "result page".

- "Unbiased Learning to Rank is proposed as an alternative and intuitive approach to utilize the user’s implicit feedback as the training signal": This is not true. It is actually proposed to mitigate the bias issue in models that rely on implicit feedback obtained from users as training signal. Implicit feedback is used in ranking models way before unbiased learning has emerged.

- "position-based biases have been extensively studied thus existing PB-ULTR algorithms may have already been adopted by the existing search engines": Does this imply that Baidu may be using a PB-ULTR algorithm in production? More importantly, are you unable to check whether the underlying Baidu dataset is the result of a PB-ULTR algorithm or not?


Minor comments:

- "the above loss is impractical": I would rather say "not scalable".

- The technique is proposed as a general solution. However, the entire presentation relies on very specific SERP features, such as multimedia type, SERP height, and ranking position.

- What is a "causal graph"? This is unclear until Page 3.

- Figures 1 and 2: These are in wrong order as Figure 2 is mentioned before Figure 1.

- Figure 2: Please separate the two subfigures properly.

- "Queries are split into 10 buckets descendingly according to the query frequency": How are they distributed to buckets (linearly, exponentially, ...)?


Corrections:
- "the real-world dataset" --> "real-world data"
- "the web with video" --> "a snippet with a video"
- "} ," --> "},"
- "\cdot" --> "\ldot"
- "confounding effect r" --> "confounding effect of r"
- "in the real-world dataset?" --> "on a real-world dataset?"
- "utilizing in" --> "utilized in"
- "to access" --> "to assess"
- "may not be necessary to" --> "does not"
- "biases" --> "biases."
- "relevant score" --> "relevance score"
- "unsatisfied results" --> "unsatisfactory results"
- "descendingly" --> "in descending order"

I acknowledge that I have read the rebuttal.

**Questions:**

Please read my review and feel free to respond to any of my concerns/comments.

**Reviewer Confidence:**

4: The reviewer is certain that the evaluation is correct and very familiar with the relevant literature

**Scope:**

4: The work is relevant to the Web and to the track, and is of broad interest to the community

---

### Official Review · Reviewer_wt4G · 2023-11-27

**Novelty:** 6
**Technical Quality:** 5

**Review:**

Bias in page presentation, particularly click behavior, hinders improving ranking models. Unbiased Learning to Rank (ULTR) algorithms address position-related bias but neglect biases from other SERP features. This paper introduces whole-page Unbiased Learning to Rank (WP-ULTR) to handle biases from all SERP features simultaneously. The proposed algorithm, called BAL, automatically discovers a user behavior model and mitigates biases using causal discovery. Experimental results validate BAL's effectiveness.

Pros:
1.This paper presents an interesting framework for automatically handling biases introduced by position, SERP features, and other factors on clicks.
2.The experimental results on the datasets are promising.
3.The theoretical work appears to be reasonable.

Cons:
1. writing has some issues. e.g. in Section 4.4, sentence "... that the curves of position-based ULTR algorithms are more distant from that of true relevance labels." is missing the comparison object.
2. some details are missing.
3. it is better to include simple models trained using human labels as baselines to see how it performs against click models.

**Questions:**

Can the model handle click biases introduced by underrepresented factors like minor race backgrounds or different body sizes?

The paragraph "Existing user behavior models [1, 23] are hand-crafted assumptions..." in Section 2 appears repetitive of the information mentioned in Section 1.

How are the initial edges of the causal graph formed? Do we require relevance labels to do so, and if yes, how many labels are needed?

The writing in Paragraph 2 of Section 3.1 is challenging to follow. What do 's,' 'i,' and 'j' represent here? Why is the relative position important, and why should the score on position 2 be larger than scores with higher positions?

More details are needed about "causal discovery," particularly for the KCI method.

Is there a typo in Section 3.2? Should it be "depend on relevance" instead of "depend on click"? I don't see any nodes depending on clicks in Figure 2.

The experiment in Figure 4(a) seems confusing. It would be helpful if the authors could clarify the writing for this experiment.

In the last paragraph of Section 4, what does "height" refer to?

**Reviewer Confidence:**

3: The reviewer is confident but not certain that the evaluation is correct

**Scope:**

4: The work is relevant to the Web and to the track, and is of broad interest to the community

---

### Official Review · Reviewer_fecm · 2023-11-29

**Novelty:** 5
**Technical Quality:** 2

**Review:**

Strengths:
+ This paper studies a new Whole-page Unbiased Learning to Rank (WP-ULTR) problem, which aims to mitigate the biases caused by all the SERP presentation features (including not only position but also multimedia type, display height, and so on).
+ This paper introduces a causal discovery method into the Unbiased Learning to Rank (ULTR) field to automatically establish the user behavior model.
+ The real-world dataset experiments demonstrate the superiority of the proposed BAL algorithm over previous ULTR models that only focus on position-related biases.

Weaknesses:
- This paper is unclear and ambiguous in the causal modeling and bias analysis of the unbiased learning-to-rank problem, which I will elaborate on later.
- Only a few SERP presentation features are included in the experiments, which is insufficient to show the superiority of the causal discovery method in modeling complex user behavior models and revealing the confounders.
- The writing of this paper needs to be improved, and I will list some specific writing issues later.

This paper formulates a new problem named Whole-page Unbiased Learning to Rank (WP-ULTR), which aims to mitigate all the SERP presentation biases instead of only those position-based biases in user click data. To solve the WP-ULTR problem, this paper first introduces a causal discovery method to model the complicated generation process of user click data and detect the SERP presentation biases using the causal graph. Then, this paper utilizes MLE to estimate the inference score on the SERP presentation features and mitigates the so-called “confounding bias” and “presentation bias” by importance reweighting and gradient blocking. The proposed BAL method shows better performance than previous position-based ULTR models on the Baidu-ULTR dataset, especially on frequent queries.

However, this paper is unclear and ambiguous in the causal modeling and bias analysis:
1>	This paper utilizes a causal discovery method to output a causal graph describing the causal relationships among “r_hat” (the relevance score from the ranking model), SERP presentation features, and the click. I understand that the causal discovery method can help automatically design the user behavior model, which is a highlight of this paper. However, my question is why you use "r_hat" instead of the true relevance "r", and I cannot find clear explanations for using "r_hat" in the paper. In the field of ULTR, we often believe that the true relevance "r" is a cause of click (for the causal graph of traditional ULTR methods, I recommend the authors refer to the paper presentation made by Chen in SIGIR’21 named "Adapting Interactional Observation Embedding for Counterfactual Learning to Rank"). It is somewhat counter-intuitive to say "r_hat" is a cause of click, so necessary justifications are needed. Moreover, if the true relevance does not even exist in the constructed user behavior model, then how can you estimate the true relevance and train an unbiased ranking model?
2>	This paper lacks rigorous theoretical analysis and proof for the so-called "confounding bias". I understand that "r_hat" is a confounder for the path “SERP feature x -> click” and can affect the correlation between SERP features and click, but I am not convinced that "this can lead to the SERP feature’s over-estimated influence on the click and relevance’s under-estimated influence on the click". The conclusions are not straightforward and require formal analysis and proof. For example, the authors can define the problem of ULTR in the language of causal inference and solve it using causal inference methods. Alternatively, they can follow Joachims et al. [23] and Harrie et al. [33] to give the exact formula of the estimator and prove its unbiasedness.
3>	Equations (8) and (9) are also confusing. In my opinion, optimizing the click probability is a common way to estimate the parameters of click models. Suppose you wish to estimate the relevance score parameters of the constructed user behavior model that describes the relationships of all the variables (including all the confounders), you can directly use MLE and gradient descent to estimate the relevance and train the ranking model since all the SERP features and click data are observable. There is no need to use importance reweighting or gradient blocking. To clarify, the previous ULTR work [23, 33] aims to directly estimate relevance from click data, so the IPW method is used to remove the bias.

Another issue of this paper is that the number of SERP features in the experiments is too small. As shown in Figure 3, only four kinds of SERP features appear in the causal graph, and two of them neither affect user click behavior nor are affected by "r_hat". Such experimental settings cannot demonstrate the superiority of the causal discovery method in modeling complex user behavior models and revealing the confounder(s). It is not hard to handcraft the causal graph if there are just four kinds of representation features.

I have two more questions. First, Line 366 states that ordinal and categorical features are converted into continuous features, which is however somewhat inconsistent with Line 480.

Second, in Line 462, why does "c" appear in the input data? "c" should not be the parent of any presentation feature.

Finally,  there are quite a few writing problems in this paper:
1>	Why is the title “learning to ranking” instead of “learning to rank”?
2>	This paper uses “SERP” to refer to “search result page presentation”, which is somewhat misleading because “SERP” often refers to “search result page” in the field of IR.
3>	Line 110, the unreasonable segmentation makes the context incoherent.
4>	Line 137, missing comma before “with the causal discovery algorithm”.
5>	Line 167, incorrect use of “other than”.
6>	Line 252, the expression “bias in the user behavior model” is inaccurate. Bias exists in the user behavior data, not the user behavior model.
7>	Figure 2, “r” appears in the caption but does not appear in the image.
8>	Figure 2, the caption says “p” refers to “position”, but the sub-figure says “p” refers to “SERP

**Questions:**

1. Why do you use "r_hat" instead of the true relevance "r"?  If the true relevance does not even exist in the constructed user behavior model, then how can you estimate the true relevance and train an unbiased ranking model?

2. Please explain why Equations (8) and (9) try to optimize a weighted likelihood of click.

**Reviewer Confidence:**

3: The reviewer is confident but not certain that the evaluation is correct

**Scope:**

4: The work is relevant to the Web and to the track, and is of broad interest to the community

---

### Decision · Program_Chairs · 2024-01-22

**Decision:**

Accept (Oral)

**Comment:**

This paper focuses on whole-page unbiased learning to rank, which attempts to mitigate biases associated with different aspects of SERP presentation.

 The reviewers appreciated the research performed and found it to be novel, but raised concerns with the causal modeling, the experimental setting, the incompleteness of the SERP presentation features, various assumptions made, the need to consider additional baselines, among other things. The majority of the concerns could be addressed with additional explanations, details (e.g., problem formalization), and reporting additional results (e.g., for ablations, etc.) already included in the rebuttal.

 The rebuttals are very thorough, although reviewer fecm appears to remain unconvinced with the authors' extensive responses to their comments. At the very least, this reviewer's comments should be addressed with care in a revision of the paper, including the continued contention over r_hat and the datasets, which was also flagged by another reviewer (fcH1).

 Overall, this paper has merits, and I would be in favor of accepting it for the conference. It is regarded by 4/5 reviewers as highly novel. The reviewer who rates this lowest in quality (2), also gives a reasonably high novelty score (5). The reviewer with the lowest novelty score (wAnD) requests a crisper definition of the problem, which I believe the authors provided in their rebuttal (and the reviewer appears to acknowledge and appreciate - along with the new ablation experiments). This should all be integrated into the paper along with the many other rebuttal comments provided by the authors to address review feedback.